# Real-time 3D single molecule tracking

Shangguo Hou [1], Jack Exell[1] & Kevin Welsher [1]✉

To date, single molecule studies have been reliant on tethering or confinement to achieve long duration and high temporal resolution measurements. Here, we present a 3D single-molecule active real-time tracking method (3D-SMART) which is capable of locking on to single fluorophores in solution for minutes at a time with photon limited temporal resolution. As a demonstration, 3D-SMART is applied to actively track single Atto 647 N fluorophores in 90% glycerol solution with an average duration of ~16 s at count rates of ~10 kHz. Active feedback tracking is further applied to single proteins and nucleic acids, directly measuring the diffusion of various lengths (99 to 1385 bp) of single DNA molecules at rates up to 10 μm$^2$/s. In addition, 3D-SMART is able to quantify the occupancy of single Spinach2 RNA aptamers and capture active transcription on single freely diffusing DNA. 3D-SMART represents a critical step towards the untethering of single molecule spectroscopy.

[1] Department of Chemistry, Duke University, Durham, NC, USA. ✉email: kevin.welsher@duke.edu

Single-molecule spectroscopy has been a critical tool in investigating molecular dynamics in biological systems[1]. Valuable insights have been gained in a wide range of sub-disciplines, ranging from DNA transcription[2–5] to enzyme catalysis[6–8], among many others. To date, the vast majority of single-molecule investigations have been restricted either to short times (using a tightly focused laser spot to monitor single molecules which transiently diffuse through the observation volume) or rely on tethering the molecule to a surface (such as TIRF or confocal single-molecule methods). The former solution-phase measurements acquire limited information due to the rapid diffusion of molecules through the focal spot. The latter tethered methods require isolation of the molecule from its environment, precluding observation of the native functions of molecules such as enzymes, which may critically rely on the crowded and non-equilibrium surroundings of the cellular interior. Several methods have sought to overcome this need to tether molecules, including liposomes[9], convex lens induced confinement (CLIC)[10,11] and the anti-Brownian electrokinetic (ABEL) trap[12–16]. Although each of these methods remove the tether, they still restrict the molecule in an isolated environment and cannot continuously monitor unbound diffusing molecules, such as an enzyme freely diffusing through the cellular interior.

A group of methods with the promise to break this tether is real-time 3D single particle tracking (RT-3D-SPT)[17]. RT-3D-SPT acquires high-speed position measurement of a diffusing particle and implements a closed feedback loop to effectively lock the rapidly moving target in the observation volume. Feedback is applied using either piezoelectric nanopositioners or galvo mirrors[18], meaning the target molecule or particle is followed at very high speed, rather than being physically confined. Pioneering work has shown that RT-3D-SPT can lock-on to a wide range of targets, from gold nanoparticles[19,20] and single quantum dots[21–23] all the way to viruses[24–26], and is easily extended to live cell tracking[27,28]. This is because the "confinement" is not a physical impediment to the target, but rather a high-speed chase which follows the particle without perturbation.

While the pioneering work on RT-3D-SPT by several groups occurred more than a decade ago, the extension of this promising method to real-time 3D single-molecule tracking (RT-3D-SMT) has been limited (Supplementary Table 1). The first reported attempt at RT-3D-SPT was by Werner and coworkers, who used a tetrahedral detection pattern to achieve real-time tracking of single Cy5-dUTP molecules in 92 wt% glycerol[27]. While truly a groundbreaking achievement for RT-3D-SMT, the trajectories were limited to several 100 ms. The same group followed-up 2 years later, showing the ability to disentangle the oligomerization states of azami green, though still in 90 wt% glycerol and still for only several 100 ms[29]. Further experiments with the same method have shown an expansion of the target scope (FRET), but with no increase in the trajectory duration despite still working in highly viscous solutions[30]. Similar results were observed by Liu and coworkers for single-labeled DNA[31].

What has limited RT-3D-SPT methods from achieving long and stable RT-3D-SMT trajectories? A commonality among the methods introduced above is the need to slow down molecular diffusion by increasing the solution viscosity for successful tracking. This is not surprising and would presumably be necessary, given that a piezoelectric nanopositioner is typically bandwidth limited to about 1 kHz (step response ~1 ms, Supplementary Fig. 2), making it impossible to track at the 100 μm²/s and greater diffusion coefficient of single molecules. For example, in the 2010 study, the use of 92 wt% glycerol slowed the diffusion of Cy5-dUTP down to ~3 μm²/s, a diffusion coefficient well within the capabilities of most RT-3D-SPT methods when tracking fluorescent beads. This means that the piezoelectric

nanopositioner should, in theory, be fast enough to follow the molecule in a 92 wt% glycerol solution. However, the duration of single-molecule trajectories was still limited despite the higher viscosity. The conclusion must be that the limited photon budget is a critical factor. This could be an issue related to either the brightness of the molecule (photons/s) or the bleaching rate (total number of photons). Starting with the latter case, it is certainly possible that these methods suffered from bleaching, leading to the short trajectories. However, it is also possible that the reduced emission rate of single molecules, compared to multiply labeled fluorescent beads, could lead to an inability to lock-on to single molecules for any appreciable period of time.

Here we break the speed limit in RT-3D-SMT by optimizing a real-time 3D tracking method for active feedback single-molecule tracking. Crossing the barrier from RT-3D-SPT to RT-3D-SMT requires knowledge of two limitations. First, the time response of the piezoelectric nanopositioner and second, the effect of limited photon information. High-speed nanopositioners are limited to a response time of about 1 ms at best (Supplementary Fig. 2). This must be taken into account when attempting to track a single molecule, even in viscous solution. Here we achieve RT-3D-SMT by starting with 3D-dynamic photon localization tracking (3D-DyPLoT, Fig. 1, Supplementary Fig. 1) and optimizing parameters with regards to (i) piezoelectric response and (ii) limited emission rate of single molecules. It is extremely difficult to find these parameters online due to the binary outcomes of tracking versus non-tracking. As such, simulations were built which could predict the ideal feedback parameters for RT-3D-SMT (Supplementary Figs. 3, 4). Once these parameters were established, they were implemented for RT-3D-SMT.

## Results

**3D-SMART.** 3D single-molecule active real-time tracking (3D-SMART) was achieved by applying these optimized parameters to the previously reported 3D-DyPLoT[25,32]. The general outline is shown in Fig. 1. A 2D electro-optic deflector and tunable acoustic gradient (TAG) lens[33] are used to rapidly scan a focused laser spot in a predefined 3D pattern around the detection area (50 kHz XY, 70 kHz Z). Photon arrival times detected by an avalanche photodiode (APD) are converted to particle positions in real-time on a field-programmable gate array (FPGA) using a Kalman update filter with assumed Gaussian density of all likelihoods[34]. The results of 3D-SMART can be seen in Fig. 2 and Supplementary Movie 1. 3D-SMART demonstrates the ability to track single Atto 647 N molecules in 90 wt% glycerol for minutes at a time, showing a two orders of magnitude increase in the tracking duration relative to previous attempts under the same conditions (See Supplementary Table 1). Figure 2a shows an example single dye trajectory. The 3D position is extracted from the piezoelectric stage coordinates as the stage is moved to maintain the fluorophore in the center of the laser scan. The single fluorophore is held in the focus of the microscope and continuously measured for >160 s (Fig. 2b). In single-molecule tracking it is important to control the molecules' concentration to avoid interference from neighboring molecules. For the excitation volume demonstrated here (1 μm in XY and 2 μm in Z), an analyte concentration of 10 pM or lower is recommended. It should be noted that the long tracking duration shown here does not rely on an oxygen scavenger system. This may be due in part to the fact that the molecule is only periodically illuminated in this scan pattern. It has been observed that the major photo-bleaching pathway in fluorophores results from photoexcitation of long-lived triplet states[35–37]. As a result, avoiding the excitation and allowing relaxation of the triplet state can reduce the rate of photobleaching. Pulse optimization to allow for dark state

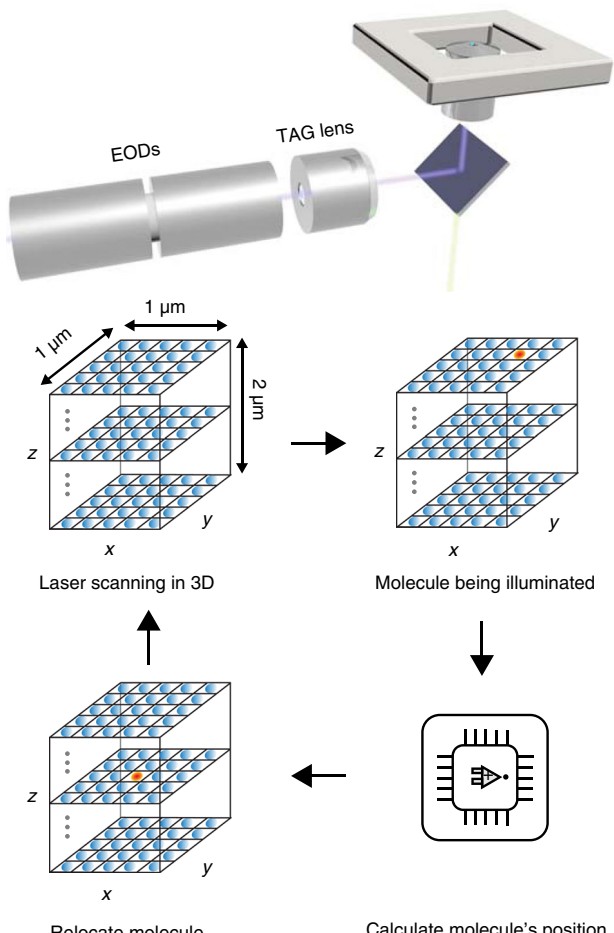

**Fig. 1 Overview of real-time 3D single-molecule tracking via 3D-SMART.**
3D-SMART is implemented using a 2D-EOD and TAG lens which rapidly
scan a focused laser spot over a small volume (1 μm × 1 μm × 2 μm, grid of
blue dots). When a molecule is illuminated (red spot), it will emit photons.
These photons are collected on an APD and the photon counts and current
laser position are used to estimate the molecule's position onboard a field-
programmable gate array (FPGA) every 20 μs. This position is then used in
a PID feedback loop to drive the piezoelectric nanopositioner to re-center
the molecule in the laser scan.

(Figure labels: 1 μm, 1 μm, 2 μm, z, x, y — "Laser scanning in 3D"; z, x, y — "Molecule being illuminated"; z, x, y — "Relocate molecule"; "Calculate molecule's position"; EODs; TAG lens)

relaxation has led to increased photon yields in both confocal
laser scanning[38,39] and STED microscopies[40,41]. The fast, periodic
illumination pattern of 3D-SMART likely reduces photobleaching
in a similar way, where the relatively long time intervals between
illumination of the tracked molecule permits triplet relaxation.
The high concentration of glycerol may also play a role, as has
been demonstrated in fluorescent proteins[42].

In any single-molecule study, it is critical to provide evidence to
support the single-molecule claim. Here we provide several pieces
of evidence to conclude that the trajectories collected are indeed
from single fluorophores. First, the motion of the molecule
displays the expected Brownian diffusion, as evidenced by a linear
mean-squared displacement (MSD). Applying the Stokes–Einstein
relation reveals a measured diffusion coefficient of 2.44 μm²/s,
corresponding to a hydrodynamic diameter of 0.98 nm, well in line
with what is expected for a single molecule (Fig. 2c, d). A further
piece of evidence comes from photon-pair correlation (PPC),
which measures the correlation of arrival times of photons on two
detectors separated by a 50/50 beamsplitter (Supplementary
Figs. 5–7). Photons coming from a single quantum emitter should
have zero probability of coincident photon detection on the two

detectors (so-called anti-bunching)[43]. An example PPC trace is
shown in Fig. 2e, demonstrating that the object tracked is indeed a
single quantum emitter. For comparison, a PPC trace taken on a
fluorescent bead at the same intensity shows a uniform PPC
(Supplementary Fig. 5c). Taken together, these two pieces of
evidence make a compelling argument that these are indeed single-
molecule trajectories.

An important consideration in any single-molecule study is the
throughput. For a method which collects serial single-molecule
traces to compile statistics, the microscope must be capable of
capturing many molecules in a row, with minimal down time
between trajectories. This is true of the 3D-SMART method.
Figure 2f shows the behavior of a solution of 4 pM Atto 647 N in
90 wt% glycerol with the feedback mechanism disabled, yielding
brief pulses of detected photons from each fluorophore. In
contrast, Fig. 2g shows the same solution with the feedback
enabled (See also Supplementary Movie 2). The microscope
demonstrates an on-time of 72.9%, meaning it spends most of its
time collecting data (Supplementary Fig. 8). Furthermore, the
observation time per molecule is extremely long, yielding an
average trajectory duration of 16.0 s (Supplementary Fig. 8),
compared to the typical burst length of a molecule randomly
diffusing through the focus of the microscope, only about 0.46 s,
even with the larger scan area of the 2D-EOD and TAG lens
(Supplementary Fig. 9). Not only does this extended time window
allow for the observation of longer timescale dynamics, but it also
increases the precision of measured molecular properties due to
the massive increase in the number of photons collected per
molecule. When feedback is disabled, a molecule diffusing
through the observation volume will yield 3400 photons on
average. When feedback is turned on, this number increases by
nearly two orders of magnitude to 193,940 photons observed for
each fluorophore (Supplementary Table 2).

**Applications of 3D-SMART**. The data presented above demon-
strate that 3D-SMART only requires a single fluorophore for
continuous observation of a molecule diffusion three dimensions.
As such, the application possibilities are expansive. Here we
demonstrate the applicability of this method to proteins, DNA,
RNA, and even multicolor dynamic processes such as DNA
transcription. Figure 3a–c shows the 3D-SMART trajectories of
single bovine serum albumin (BSA) labeled non-specifically with
NHS-Atto 647 N in 73.5 wt% glycerol. Given the larger size of the
protein compared to a single fluorophore, the concentration of
glycerol can be reduced without a sacrifice in the tracking ability.
Additional support for the single-molecule hypothesis is observed
in these single BSA trajectories. Since the BSA are non-specifically
labeled, a portion of the proteins are doubly labeled. This can be
seen clearly in Supplementary Fig. 10 and Supplementary Movie 3,
where during the course of the trajectory one of the fluorophores
blinks on and off. During this time, the intensity doubles, showing
that the intensity observed above for single Atto 647 N above (~10
kHz) does indeed correspond to the single-molecule intensity
level. Similar results for BSA were observed when using a different
label (Alexa 488, Supplementary Figs. 11, 12).

Applying 3D-SMART to track double-strand DNA (dsDNA)
in solution at high temporal resolution for long times enables
examination of different models for the diffusion of semiflexible
polymers. By tracking different lengths of dsDNA (99 bp in
70 wt% and 40 wt%, 547 bp in 25 wt% glycerol, and 1385 bp in
PBS, Fig. 3d–g, Supplementary Figs. 13, 14), we were able to use
3D-SMART to test the relative accuracy of the Rouse (which
ignores the excluded volume and hydrodynamic interactions) and
Zimm (which adds a hydrodynamic interaction) model for
diffusion of a flexible polymer[44] as compared to a more

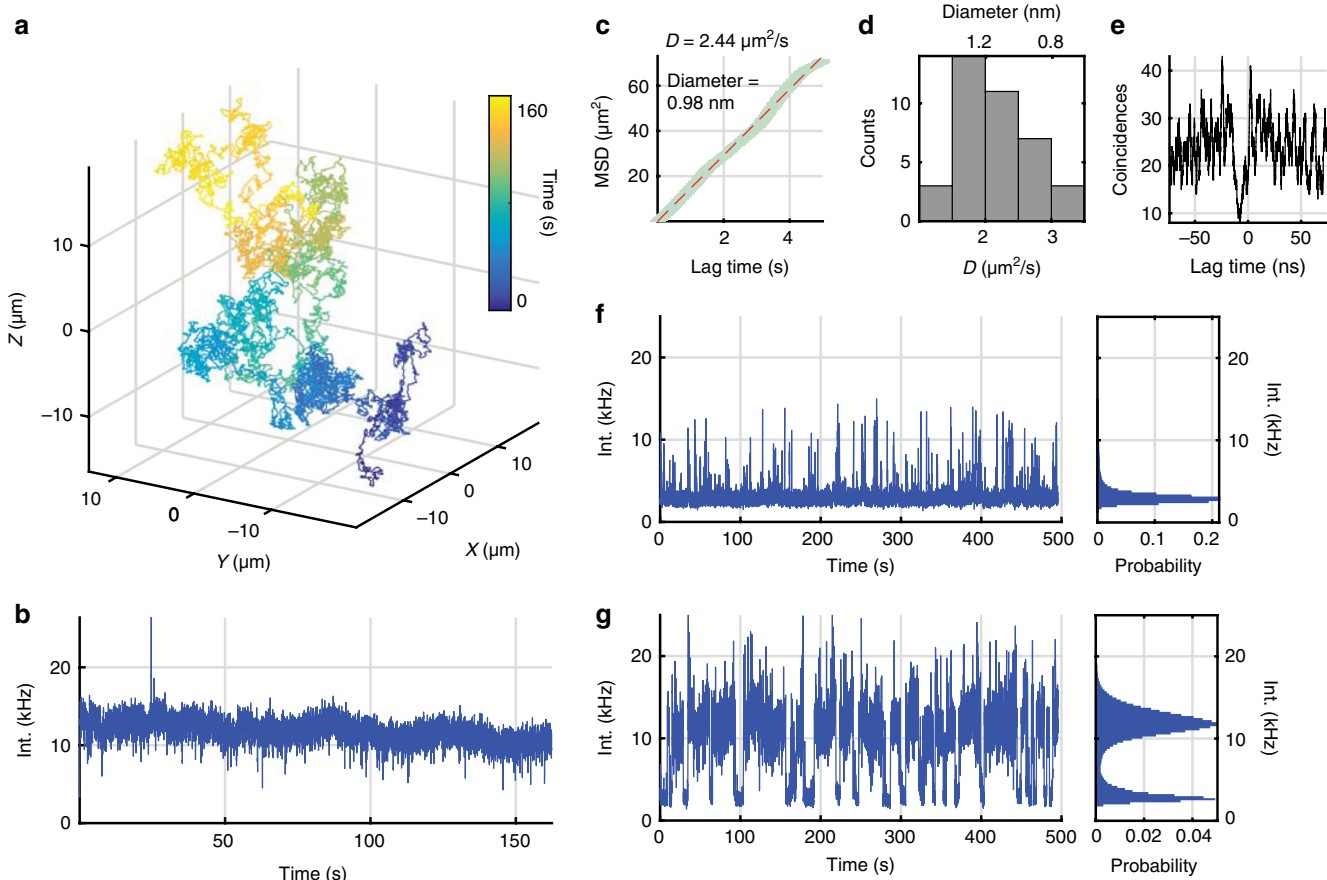

**Fig. 2 Locking on to single fluorophores with 3D-SMART. a** 3D trajectory of a single Atto 647 N molecule tracked for 160 s in 90 wt% glycerol. XYZ coordinates are extracted from the readout of the piezoelectric nanopositioner during active feedback. **b** Intensity trace recorded during the trajectory shown in (**a**), showing uniform intensity despite the rapid diffusion of the molecule. **c** MSD versus lag time and linear fit of the trajectory in (**a**), yielding a hydrodynamic diameter of 0.98 nm, consistent with a single fluorophore. **d** Histogram of diffusion coefficients and hydrodynamic diameters of a population of Atto 647 N trajectories (number of trajectories $n = 38$). **e** Photon-pair correlation trace for the molecule in Fig. S6a, showing anti-bunching of photon arrivals at short lag time. **f, g** Comparison of single-molecule data collection with feedback (**f**) off and (**g**) on. Histograms demonstrate the increase in photon collection with the feedback enabled, as well as the uniform observed emission rate from tracked fluorophores.

sophisticated wormlike chain model which takes chain stiffness into account[45]. We found that the 3D diffusion of the various lengths of dsDNA, as measured by 3D-SMART, did not conform to the simple model of a flexible polymer put forth by Rouse and Zimm, but agreed almost exactly with the WLC model developed by Yamakawa and Fujii (Supplementary Figs. 15, 16). For comparison, the data from the WLC chain model as well as 3D-SMART tracking data were in agreement with light-scattering measurements of DNA restriction fragments[46]. These data, taken together, suggest the lack of consideration of chain rigidity in the Rouse and Zimm model leads to underestimation of the diffusion coefficient for DNA of a given length.

Furthermore, dsDNA tracking is useful for extrapolating the speed and size limits of 3D-SMART. For single dye molecules, the required glycerol concentration for successful active feedback tracking is 90 wt% glycerol. For single proteins (BSA), it is 73.5 wt% glycerol. For dsDNA at a length of 1385 bp, with a single fluorophore labeled at the terminus, active feedback tracking is achieved in PBS (0 wt% glycerol, Supplementary Movie 4), with diffusion coefficients up to ~7 μm²/s. While it has been found that glycerol can have an effect on DNA conformation[47], our data suggest that shorter DNA still fits well to the WLC model. This is evidenced by the same effective diffusion coefficient measured for 99 bp dsDNA in 70 wt% and 40 wt% glycerol. At lower glycerol concentrations (25 wt%), longer

dsDNA (547 bp) still shows good agreement with the WLC model, indicating that no change in the dsDNA conformation as function of glycerol concentration can be detected within the resolution of these measurements. This demonstrates that single fluorophore tracking is achievable in low viscosity environments at high diffusive speeds.

A further application of 3D-SMART is to analyze the properties of distributions of single molecules. While a bulk measurement will yield an average value, the only way to truly extract the molecular distribution is to use single-molecule methods. 3D-SMART enables extraction of single-molecule distributions in the solution phase. As a demonstration, 3D-SMART was used to track Spinach2 RNA aptamers[48,49]. The Spinach2 aptamer binds to small molecule chromophores (DFHBI-1T, among others). Upon binding, the chromophores are stabilized by the aptamer and go from nonfluorescent to fluorescent. By tracking individual aptamers, the number of bound chromophores can be extracted (once the intensity level of a single chromophore is established via single-step bleach, Fig. 3j, Supplementary Fig. 18). Using 3D-SMART, it was observed that each aptamer was labeled with $7.09 \pm 6.46$ (mean ± s.d.) DFHBI-1T molecules, compared to the expected value of 8 based on the number of repeats in the sequence (Fig. 3k, Supplementary Movie 5). However, examination of the actual distribution shows a number of aptamers with much larger

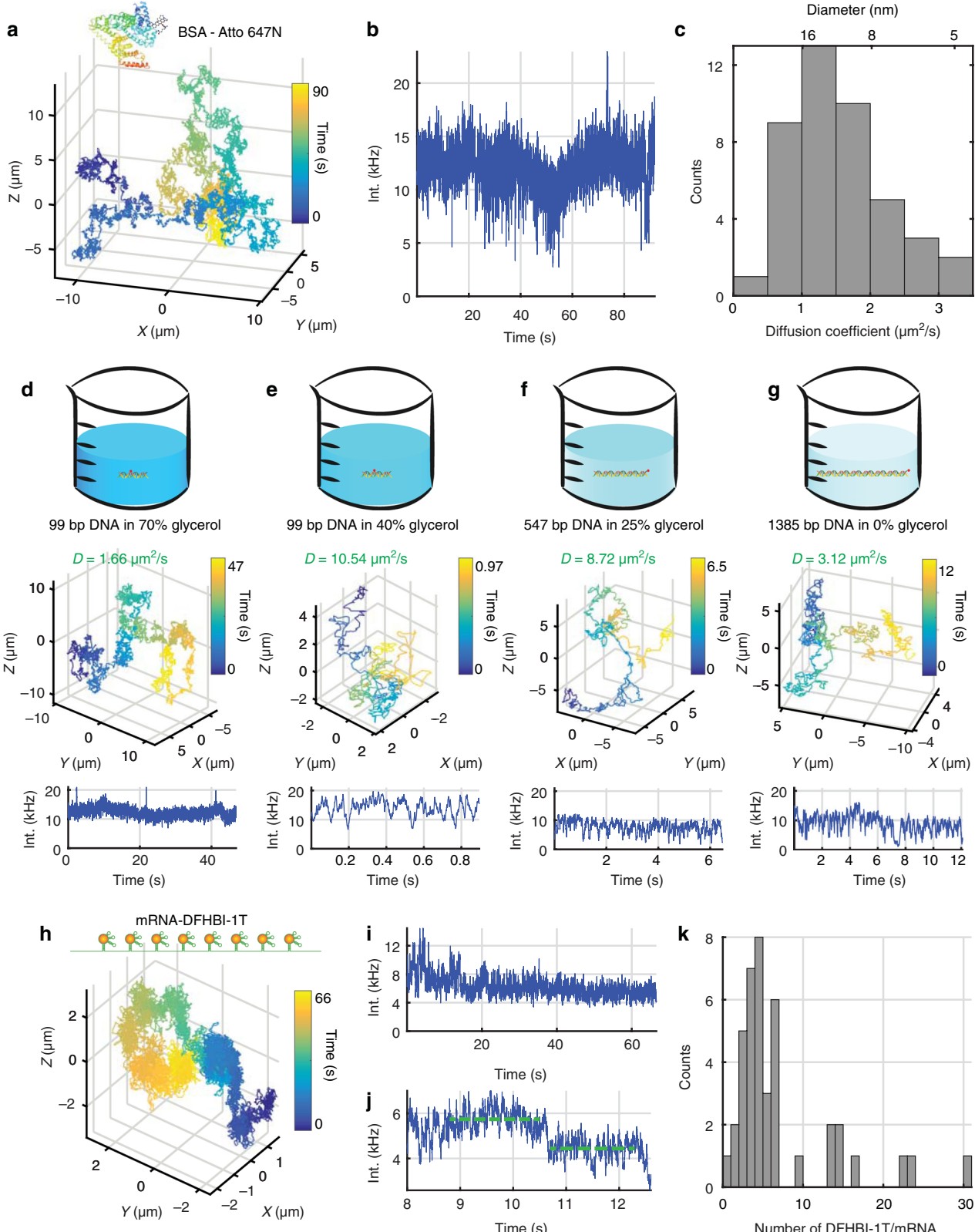

**Fig. 3 Application of 3D-SMART to biomolecules. a–c**, 3D trajectory, intensity trace, and histogram of diffusion coefficients for single BSA molecules in 73.5 wt% glycerol (number of trajectories $n = 43$, bin siz $= 0.5 \ \mu m^2/s$). **d–g** Tracking a range of dsDNA lengths at indicated glycerol concentrations. **d** 99 bp dsDNA tracking in 70 wt% glycerol-water solution. **e** 99 bp dsDNA tracking in 40 wt% glycerol. **f** 547 bps dsDNA tracking in 25 wt% glycerol. **g** 1385 bp dsDNA tracking in PBS solution. **h, i** 3D trajectory and intensity trace of DFHBI-1T labeled single Spinach RNA aptamers tracking in 75 wt% glycerol. **j** Stepwise bleaching is used to calculate the intensity of single DFHBI-1T dye. **k** The histogram of the number of DFHBI-1T for single Spinach RNA aptamers (number of trajectories $n = 41$, bin size $= 1$).

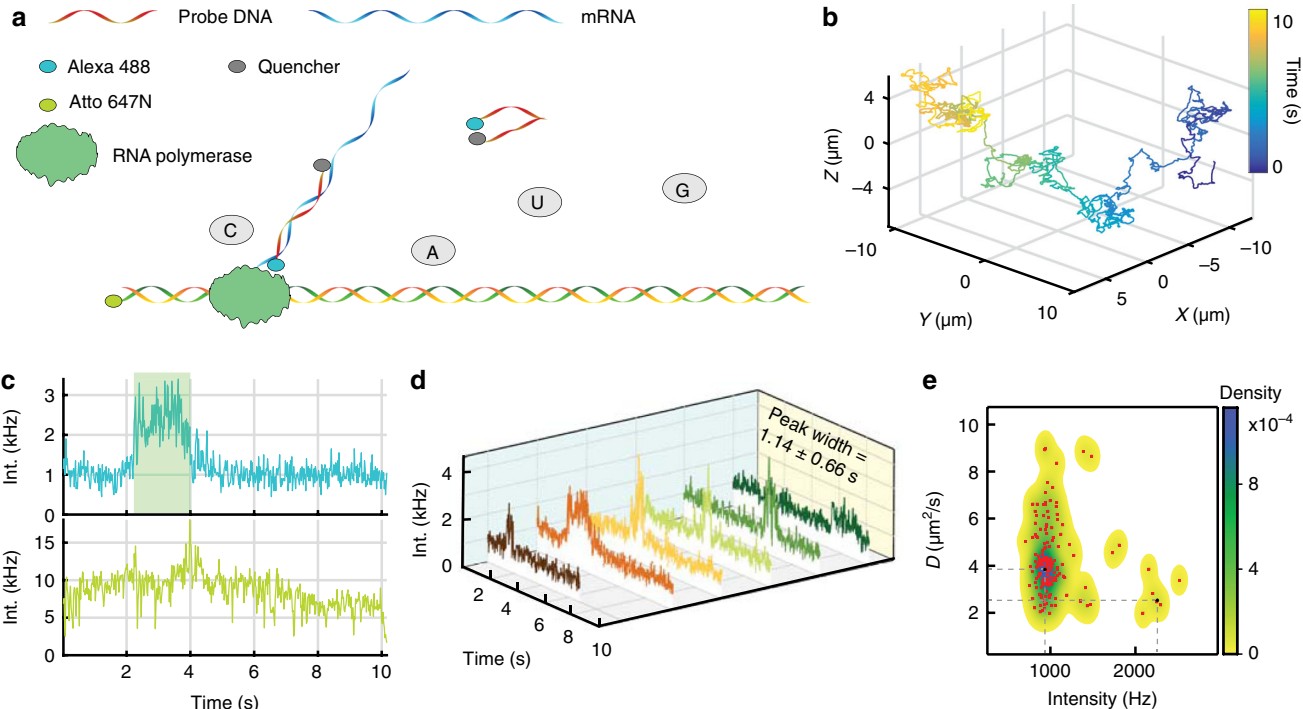

**Fig. 4 Real-time measurement of transcription on single, freely diffusing dsDNA. a** Solution-phase single-molecule transcription is performed by labeling 1218 bp DNA with Atto 647 N and adding rNTPs and *E.coli* RNA polymerase in 22 wt% glycerol. The transcribed RNA is sensed using a complementary ssDNA with Alexa 488-Iowa Black fluorophore-quencher pair. The ssDNA is quenched in its unpaired state and becomes fluorescent upon pairing with RNA. **b** 3D trajectory of dsDNA in transcription buffer. **c** Intensity of dsDNA from (**b**) (bottom) and the observed transcription burst from the ssDNA probe (top, highlighted with light green shadow). **d** Six transcription bursts from the ssDNA probe produce an average burst duration of 1.14 ± 0.66 s. **e** 2D scatter plot of photon counts of mRNA channel versus diffusion coefficient of template double strands DNA. 6 trajectories in (**d**) are used to produce this plot. The underlying density map is visualized using 2D kernel density estimation. Density <10⁻⁵ is shown as white. Two density peaks (corresponding to the inactive and active transcription states) are marked with the black dot and dotted gray lines. The higher intensity spots (to the right) correspond to active transcription state while lower intensity spots (to the left) correspond to the inactive transcription state. The photon counts and diffusion coefficient are calculated with 500 ms bin time.

numbers of chromophores (likely due to aggregation), with a more normal distribution peaked around 5 DFHBI-1T per aptamer. Interestingly, zero aptamers were observed with a full 8 chromophores.

As a final demonstration of the power of 3D-SMART, we show the observation of a compete biochemical reaction at the single-molecule level in solution (Fig. 4). To do so, 1218 bp dsDNA with a single Atto 647 N label was mixed with NTPs, *E. coli* RNA polymerase and complementary ssDNA with a fluorophore-quencher (Alexa Fluor 488 - Iowa Black RQ) pair to act as a fluorescence in situ hybridization (FISH) marker. Upon transcription of the DNA by the RNA polymerases, the FISH probe complements to the nascently transcribed RNA, separating the fluorophore from the quencher and resulting in a burst of fluorescence (Supplementary Fig. 19). Using 3D-SMART, bursts of fluorescence, with an average of 1.14 ± 0.66 s in duration, can be observed from single DNA molecules being transcribed while undergoing free Brownian motion in three dimensions. The fastFISH probe DNA sequence used here is the same as the original implementation[50], and its hybridization rate is $6 \times 10^6\,M^{-1}\,s^{-1}$ without the presence of glycerol. The reported mRNA transcription rate of *E.coli* RNAP varies from 10 to 36 nt s$^{-1}$ [51–53] and the produced mRNA length in here is 90 nt, which only opens a very short time window for probe DNA to hybridize with the transcribed RNA. The probe DNA concentration was kept low (200 pM) to reduce fluorescence background, resulting in a low transcription detection rate in these experiments (6 of 628

trajectories) since hybridization rate is only $1.2 \times 10^{-3}\,s^{-1}$ without the presence of glycerol. Moreover, the presence of glycerol will further decrease the hybridization kinetics[54,55]. Considering the low probe binding rate in the experiment, the start points of the transcription signal in Fig. 4c, d likely do not indicate the start of transcription, rather[54] the binding of the fastFISH probe after the transcription has already started. On the other hand, the Alexa 488 bleaching time (the 99% confidence level: 17.7 ± 12.4 s, Supplementary Fig. 20) is significantly longer than the duration of the transcription signal (~1 s, $p = 0.0091$ using a one-sided *t*-test), so the disappearance of the signal should indicate dissociation of the transcribed RNA from the diffusing dsDNA. The results of this assay can be seen in Fig. 4 (Supplementary Movie 6). It should be noted that there was some tendency for DNA molecules to aggregate in the solution, and these trajectories where eliminated by thresholding out trajectories which exhibited intensities larger than that of a single dye (Supplementary Fig. 21). Importantly, the untethering of this fastFISH measurement not only removes the possibility of surface effects but also adds a new dimension of analysis. Examining the relationship between the ssDNA signal and the diffusion of the target dsDNA reveals that there is a drop in the diffusion coincident with the transcription event (from D ~4 µm²/s to D ~2.5 µm²/s, Fig. 4e, Supplementary Fig. 22), indicative of binding between the dsDNA and the polymerase, a distinction impossible to make with a surface bound measurement. Control experiments preformed without adding *E.coli*

RNAP resulted with no transcription bursts associated with probe binding observed.

## Discussion

The above demonstration of 3D-SMART shows a dramatic improvement in tracking speed and information acquired from single molecules in solution, enabling continuous observation of a single, freely diffusing dye molecule for up to minutes at a time and diffusion coefficients up to 10 $\mu m^2$/s, all with photon-limited temporal resolution. 3D-SMART has the power to completely untether single-molecule experiments. This opens the path towards measuring the fast internal molecular motions of single proteins in the complex cytoplasmic milieu, though this represents only a tiny portion of what 3D-SMART can enable. One reason single-molecule observation has been so powerful in biological systems is that molecules are intrinsically "confined" to the cellular container, making observation over long times more feasible. 3D-SMART can extend single-molecule spectroscopy to completely unconfined systems, going beyond molecular biology to synthetic chemistry (e.g., living polymerization[56]) and condensed matter physics (e.g., reptation of single polymers in heterogenous systems). Untethering single-molecule spectroscopy in this way is critical towards increasing the application scope of these powerful methods.

## Methods

**3D-SMART setup design.** The 3D-SMART setup uses a 488 nm frequency-doubled solid-state laser (FCD488-30, JDSU) and 640 nm diode laser (OBIS 640LX, Coherent) for excitation. A telecentric system, consisting of two lenses with focal length $f = 150$ mm (AC254-150-A-ML, Thorlabs) and $f = 200$ mm (AC254-200-A-ML, Thorlabs), and a pinhole with 75 μm diameters (P75S, Thorlabs) are used to collimate and purify the 488 nm beam. The polarization state of 488 nm beam is cleaned by a Glan-Thompson Polarizer (GTH5-A, Thorlabs) and tuned by a half-wave plate (WPH05M-488, Thorlabs). For 640 nm laser beam, two lenses with focal length $f = 100$ mm (AC254-100-A-ML, Thorlabs) and $f = 150$ mm (AC254-150-A-ML, Thorlabs) and a pinhole with 100 μm diameters (P100S, Thorlabs) are used for collimation. The polarization state of the 640 nm beam was cleaned by a polarizer (WP25M-VIS, Thorlabs) and tuned by a half-wave plate (WPH05M-633, Thorlabs). A dichroic mirror (FF614-SDi01, Semrock) is used to combine the 488-nm beam and 640-nm beam. The beams then travel through two electro-optic deflectors (EOD; M310A, ConOptics). The EODs are used to produce the designed scanning pattern. A lens pair (AC254-75-A-ML; AC254-250-A-ML, Thorlabs) is placed after the EOD to act as beam expander. The TAG lens (TAG Lens 2.5, TAG optics) is used to modulate the beams focal position dynamically. The TAG lens has a dynamically changing focus with low aberration performance and up to megahertz frequencies. Here we run the TAG lens at the ~70 kHz resonance. The beam from the TAG lens is relayed to the microscope objective lens (×100, 1.49 NA, M27, Zeiss) by two $f = 200$ mm focal length lenses (AC254-200-A-ML, Thorlabs). A dichroic mirror (ZT405/488/561/640rpc, Chroma) reflects the beam towards the back aperture of the objective lens. A bandpass filter (for Atto 647N, ET706/95m, Chroma; for Alexa 488 and DFHBI-1T fluorescence, ET535/50m, Chroma) separates the fluorescence emission from the excitation lasers. For two color experiments, a dichroic filter (T610lpxr, Chroma) separates the fluorescence onto different detection paths. Two APD are used to measure fluorescence signal with a lens with focal length 50 mm (AC254-50-A-ML, Thorlabs) installed before each APD. A micropositioning system (Microstage, MicroStage Series, Mad City Labs) enables coarse sample positioning. A 3D piezoelectric stage system (XY—Nano-PDQ275HS, Z—Nano-OP65HS, Mad City Labs) is used to move the sample and objective lens for active feedback 3D tracking. In photon antibunching experiment, a 50/50 beamsplitter (21000, Chroma) was used to split Atto 647N fluorescence onto the two APDs and a time-correlated single-photon counting model (TimeHarp 260, PicoQuant) tags the arrival times of fluorescence photons in each APD. An IR blocking filter (FF01-715/SP-25, Semrock) is placed in front of one APD for preventing optical cross talk between two detectors.

A field-programmable gate array (FPGA, NI-7852r National Instruments) based data-processing unit controls the EOD scanning, collects photon counts from the APD, interpolates the TAG lens signal, calculates the particle position estimates, applies position feedback to the piezoelectric stage, and records the particle position information. The system operates in custom software implemented with LabView 15 (National Instruments).

For Alexa 488 tracking, the laser power was ~1.7 μW at the objective focus. For Atto 647 N tracking, the laser power was ~2 μW. For mRNA-DFHBI-1T tracking, the laser power was 1.9 μW. All the tracking experiments were performed at room temperature (~23 °C).

**Sample preparation for single dye tracking.** Atto 647 N NHS ester (18373, sigma) or Alexa 488 (A20100, Invitrogen) was first dissolved in Dimethyl Sulfoxide (DMSO, VWRVN182, VWR) at 5 mg/ml and aliquoted into 5 μl for each vial and stored in −80 °C. Then the dye was diluted into 10 pM with distilled water. Then 1 mL dye solution was mixed with 9 g glycerol (15514011, Invitrogen) and added 400 μl solution to a home-made sample holder for 3D-SMART experiments (Supplementary Fig. 23).

**Sample preparation for BSA tracking.** Bovine serum albumin (BSA, A2153, Sigma) was dissolved in DPBS (14190144, Gibco) at 2.6 mg/ml (40 μM). Then 40 μl BSA was mixed with 1 μL Atto 647 N NHS ester or Alexa 488 NHS ester (5 mg/ml). The mixture was incubated for 2 h (25 °C) on shaking stage. The labeled BSA solution was then purified with Zeba spin desalting columns (89882, Thermo Scientific) according to the manufacturer's procedure. The purified BSA solution was diluted to 2–5 pM in 73.5 wt% glycerol-PBS solution. 400 μL of the solution was added to the home-made sample holder for 3D-SMART experiments.

**DNA sequences and labeling.** To tag the DNAs with fluorescent molecules, DNAs were modified with amino (synthesized by Integrated DNA Technologies). To label the amino modified DNA with Atto 647 N NHS or Alexa 488 NHS, the amino modified DNA was dissolved in PBS at a concentration of 100 μM, then incubated with dye at a molar ratio of ~1:5 for 2 h (25 °C) while shaking. The labeled dsDNA was then purified with NAP-5 G25 column (17085301, GE Healthcare illustra) according to the product manual. All the amino modified DNA premers and single DNA strand were labeled with fluorescent molecules before the PCR amplification and hybridization.

**99 bp dsDNA.** For 99 bp dsDNA tracking the following sequence was used (synthesized by Integrated DNA Technologies): 5′-TAT GTA ATT GGA GTG GTT AAG ATA AGG GAT AGG GTG AAA TTG TTA T /amino/ C CGC TCT CAC AAT TCC ACA CAT TAT ACG AGC CGA AGC ATA AAG TGT CAA GCC T-3′. Two complementary DNAs were mixed in PBS buffer with a concentration of 1 μM and then went through the following annealing procedure: (1) heat to 95 °C and maintain the temperature for 5 min; (2) Cool to 4 °C maintain the temperature for temporary storage.

**547 bp dsDNA.** For 547 bp dsDNA tracking, the following sequence was used: 5′-CCACAACGGTTTCCCTCTAGAAATAATTTTGTTTAACTTTAAGAAGGAGA TATACATAaaagacgccttgttgttagccataaagtgataacctttaatcattgtctttattaatacaactcactataa ggagagacaacttaaagagacttaaaagattaatttaaaatttatcaaaaagagtattgacttaaagtctaacctataggat acttacagccatgtagtaaggaggttctaatagccatcccaatcgacaCCCTATCCCTTATCTTAACCA CTCCAATTACATACACCTTTCAAAACTTCAAACCTTTTAGATACCTGATA CAAAGTCCATTATGATTTTTAGATTTCGTATATTTACACTTGCACCATAC GCATGTAAAATTAGAAGCAAAGTACGATTCTTAGACCGTATGTATAATAT AATTATGTAGATGTGATGAGTTTCTTTTATATGCTTCACCTGTCGGATCG GTCTGCAGCTGGATATTACGGCCTTTTTAAAGACCGTAAAGAAAAATAA GCACAAGTTTTATCCGGAA-3′. Here, lower case letters indicate the T7A1 promoter region (−163 to +38 from the transcription start site represented by the bold typed a). Region −221 to +91 is the same template sequence as previously reported (42), except for a small alteration in the initial transcribed region (ITR). Region −200 to +325 containing the T7A1 promoter region and the probe binding site was synthesized by Integrated DNA Technologies (IDT) as double-stranded DNA delivered on a pUCIDT plasmid. The sequence was then excised with XbaI and HindIII and inserted into pT7-7 plasmid (Addgene #36046) between the XbaI and HindIII restriction sites to generate a plasmid template construct (pT7-fast-FISH (pT7-fF)).

The 547 bp DNA template was amplified by PCR from pT7-fF using the following primers:
Primer 1: 5′-CCACAACGGTTTCCCTCTAG-3′
Primer 2: 5′-/Amino/TTCCGGATAAAACTTGTGC-3′

**1385 bp dsDNA.** For 1385 bp dsDNA tracking and DFHBI-1T-mRNA tracking experiments, the following sequencing containing eight spinach2[40] sites was used: 5′-CCACAACGGTTTCCCTCTAGAAATAATTTTGTTTAACTTTAAGAAGGA GATATACATAaaagacgccttgttgttagccataaagtgataacctttaatcattgtctttattaatacaactcacta taaggagagacaacttaaagagacttaaaagattaatttaaaatttatcaaaaagagtattgacttaaagtctaacctataggatact tacagccatgtagtaaggaggttctaatagccatcccaatcgacaCCCTATCCCTTATCTTAACCACTC CAATTACATACACCTTTCAAAACTTCAAATTCACGTAAGATGCTCCGGTTA GGGATCTAGATAGACGGCATGGGGAGATGTAACTGAATGAAATGGTGAA GGACGGGTCCAGTAGGCTGCTTCGGCAGCCTACTTGTTGAGTAGAGTGTG AGCTCCGTAACTAGTTACATCACTGATGTACCGTTGAGCAGGGAGATGT AACTGAATGAAATGGTGAAGGACGGGTCCAGTAGGCTGCTTCGGCAGCC TACTTGTTGAGTAGAGTGTGAGCTCCGTAACTAGTTACATCACTCGCTAG AGCATGGTTTGGGAGCTAGATAGACGGCATGGGGAGATGTAACTGAATG AAATGGTGAAGGACGGGTCCAGTAGGCTGCTTCGGCAGCCTACTTGTTG AGTAGAGTGTGAGCTCCGTAACTAGTTACATCACTGATGTACCGTTGAGC AGGGAGATGTAACTGAATGAAATGGTGAAGGACGGGTCCAGTAGGCTGCT TCGGCAGCCTACTTGTTGAGTAGAGTGTGAGCTCCGTAACTAGTTACATC

ACTCGCTAGAGCATGGTTTGGGAGCTAGATAGACGGCATGGGGAGATG
TAACTGAATGAAATGGTGAAGGACGGGTCCAGTAGGCTGCTTCGGCAGC
CTACTTGTTGAGTAGAGTGTGAGCTCCGTAACTAGTTACATCACTGAT
GTACCGTTGAGCAGGGAGATGTAACTGAATGAAATGGTGAAGGACGG
GTCCAGTAGGCTGCTTCGGCAGCCTACTTGTTGAGTAGAGTGTGAGCT
CCGTAACTAGTTACATCACTGATGTACCGTTGAGCAGGGAGATGTAACT
GAATGAAATGGTGAAGGACGGGTCCAGTAGGCTGCTTCGGCAGCCTAC
TTGTTGAGTAGAGTGTGAGCTCCGTAACTAGTTACATCACTGATGTAGAGA
CGGCATGGGGAGATGTAACTGAATGAAATGGTGAAGGACGGGTCCAGT
AGGCTGCTTCGGCAGCCTACTTGTTGAGTAGAGTGTGAGCTCCGTAACT
AGTTACATCACTGATGTACCGTTGAGCAGGGAGATGTAACTGAATGAAAT
GGTGAAGGACGGGTCCAGTAGGCTGCTTCGGCAGCCTACTTGTTGAGTA
GAGTGTGAGCTCCGTAACTAGTTACATCACTCGCTAGAGCATGGTTTGGG
AGCTAGCGCACAAGTTTTATCCGGAA-3′. Here, lower case letters indicate the
T7A1 promoter region (−49 to +38 from the transcription start site represented by
the bold typed a is identical to the T7A1 promoter region in pT7-fF). The underlined
portion indicates the "Spinach 2" aptamer sequence[40]. Region +95 to +374 con-
taining (linker-Spinach 2-linker-Spinach 2-linker) was synthesized by GenScript as
double-stranded DNA delivered on a pUC57 plasmid. This construct was inserted
between the probe-binding site and HindIII restriction site of pT7-fF to create pT7-
Spi2-2R. The final plasmid template construct pT7-Spi2-8R, containing eight "Spi-
nach 2" aptamers, was created using the repeat expansion method[57]. The 1385 bp
DNA transcription template was amplified by PCR from pT7-Spi2-8R using the
Primer 1 and Primer 2.

**Transcription sequences (1218 bp dsDNA).** For ensemble and single-molecule
diffusion fastFISH experiments, the following sequence containing the single
probe-binding site was used:

5′-GCTAATCCTGTTACCAGTGGCTGCTGCCAGTGGCGATAAGTCGTGT
CTTACCGGGTTGGACTCAAGACGATAGTTACCGGATAAGGCGCAGCGGT
CGGGCTGAACGGGGGGGTTCGTGCACACAGCCCAGCTTGGAGCGAACGA
CCTACACCGAACTGAGATACCTACAGCGTGAGCATTGAGAAAGCGCCAC
GCTTCCCGAAGGGAGAAAGGCGGACAGGTATCCGGTAAGCGGCAGGGT
CGGAACAGGAGAGCGCACGAGGGAGCTTCCAGGGGGAAACGCCTGGTA
TCTTTATAGTCCTGTCGGGTTTCGCCACCTCTGACTTGAGCGTCGATTTT
TGTGATGCTCGTCAGGGGGGCGGAGCCTATGGAAAAACGCCAGCAACG
CGGCCTTTTTACGGTTCCTGGCCTTTTGCTGGCCTTTTGCTCACATGTTC
TTTCCTGCGTTATCCCCTGATTCTGTGGATAACCGTATTACCGCCTTTGA
GTGAGCTGATACCGCTCGCCGCAGCCGAACGACCGAGCGCAGCGAGTCA
GTGAGCGAGGAAGCGGAAGAGCGCCTGATGCGGTATTTTCTCCTTACGC
ATCGTGTCGGTATTTCACACCGCATACAGATCTGTATGGTGCACTCTCAG
TACAATCTGCTCTGATGCCGCATAGTTAAGCCAGTATATACACTCCGCTA
TCGCTACGTGACTGGGTCATGGCTGCGCCCCGACACCCGCCAACACCCG
CTGACGCGCCCTGACGGGCTTGTCTGCTCCCGGCATCCGCTTACAGACA
AGCTGTGACCGTCTCCGGGAGCTGCATGTGTCAGAGGTTTCAACGCTCA
TCACCGAAACGCGCGAGGCCCAGCGATTCGAACTTCTGATAGACTTCGA
AATTAATACGACTCACTATAGGGAGACCACAACGGTTTCCCTCTAGAAA
TAATTTTGTTTAACTTTAAGAAGGAGATATACATAaaagacgccttgttgttagccataa
agtgataacctttaatcattgtctttattaataacaactcactataaggagagacaacttaaagagactaaaagattaatttaa
aaatttatcaaaaagagtattgacttaaagtctaacctataggatacttacagccatgtagtaaggaggttctaatagccat
cccaatcgacaCCCTATCCCTTATCTTAACCACTCCAATTACATACACCTTTCA
AAACTTCAAA-3′. The 1218 bp DNA transcription template was amplified by
PCR from pT7-fF using the following primers:

Primer 3: 5′-GCTAATCCTGTTACCAGTGG-3′
Primer 4: 5′-/amino/TTTGAAGTTTTGAAAGGTGTATG-3′

As above, lower case letters indicate the T7A1 promoter region (−163 to +38
from the transcription start site represented by the bold typed **a**). Here, the
underlined portion indicates the oligo probe-binding site. The self-quenched oligo
probe is 5′-/Alexa488/GTTAAGATAAGGGATAGGG/RQ/-3′. RQ indicates Iowa
Black RQ.

All the primers and self-quenched probe were both synthesized by synthesized by
Integrated DNA Technologies and primer 2 and primer 4 containing a C6-amino to
allow coupling to ATTO647N-NHS. The dsDNA strands produced by PCR, 547 bp,
1218 bp, and 1385 bp, were purified using a PCR purification kit (Qiagen).

**dsDNA sample preparation for tracking.** For 1385 base pairs DNA tracking in
PBS, the DNA was firstly dissolved in PBS and incubated for 30 min. For 99 and
546 bp DNA tracking, 70, 40 or 25 glycerol (by weight) was added after incubating
the dsDNA in PBS for 30 min or by going through the annealing procedure as
described above. The sample was then further diluted for tracking via 3D-SMART.
The final concentration of DNA used in this work was 1–5 pM.

**DFHBI-1T-mRNA tracking.** DFHBI-1T (170629, Lucerna) was dissolved in DMSO
at 31 mM concentration. Before mixing the DFHBI-1T with mRNA, the DFHBI-1T
was diluted to 0.31 mM with DI water and titrated to neutral pH with KOH. Then
1.2 k bps mRNA (200 nM) was incubated with DFHBI-1T (10 μM) for 1 h in
binding buffer (40 mM HEPES (PH 7.2–7.5), 125 mM KCl, 10 mM MgCl₂). The
tracking solution was made by adding 30 μL of the mixture to 750 mg glycerol,
200 μL 5X binding buffer and 20 μL H₂O.

**DFHBI-1T counting in single Spinach RNA aptamers.** The number is calculated
by $n = (I_{mRNA} - background) \times I_{DFHBI}^{-1}$, in which $I_{mRNA}$ is the mean intensity of
first 100 ms trajectory and $I_{DFHBI}$ is the single DFHBI-1T intensity. Single DFHBI-
1T fluorescence intensity is derived from the step-wise bleaching intensity. The
mean of three stepwise bleaching steps from three trajectories was measured to be
1323 ± 93 Hz.

**Ensemble RNA transcription.** To validate transcription and oligo probe bind-
ing, bulk experiments were performed in buffer containing 40 mM Tris-HCl
pH = 8.0, 25 mM MgCl₂, 2.5 mM rNTPs (GTP, UTP, CTP, ATP), 20% (v/v)
glycerol, 2.5 mM spermidine, 0.01% Triton X-100, 0.05 units YIPP (NEB
#M2403S), 1 unit RNaseOUT (#10777019,Invitrogen), 1 mM DTT, and 200 ng
of the 1218 bp DNA template. 2.5 units of T7 RNA Polymerase (New England
Biolabs #M0251S) was added to initiate the reaction, which was then
allowed to proceed at 37 °C for 2 h. To this reaction mixture the self-
quenched oligo probe was added at a final concentration of 1.5 μM. The mixture
was incubated at 37 °C for 30 min prior to acquiring fluorescent emission
spectra.

**DNA transcription tracking.** All the in vitro transcription experiments were
performed in NEB's transcription buffer (40 mM Tris-HCl, 6 mM MgCl2, 1 mM
DTT, 2 mM spermidine) with 1 mM NTPs (R1481, Thermo Scientific), 22 wt%
glycerol, 100 U E.coli RNAP (M0551S, NEB), 200 pM probe DNA, 80 U RNase
Ribonuclease Inhibitor (10777019, Invitrogen) and 12 pM template DNA. Before
the transcription experiment, the double-strand template DNA was diluted into
PBS with a concentration of 6 nM and incubated for 30 min in room temperature.
Hundred and ten milligrams of glycerol was mixed with 314 μL DI water and 50 μL
10x transcription buffer. The sequentially added NTPs, RNase Ribonuclease
Inhibitor, template DNA, RNAP and probe DNA. For in vitro transcription control
experiment, all the experiment conditions were same as transcription experiments
except without E.coli RNAP. All these experiments were performed in room
temperature (23 °C).

**Identification of single transcription events.** The transcription bursts are
determined by inspection. As can be seen Fig. 4c, the background level is ~1 kHz
and the intensity increases to ~2–2.5 kHz when the probe binds. We define a burst
when the intensity is greater than 150% of the background (here 1.5 kHz). The
burst width is measured from the full-width at half-max. In the transcription
experiment, trajectories for which the signal of the DNA channel or mRNA
channel exceeded the single-molecule level intensity (>20 kHz for DNA channel
and >4 kHz for mRNA channel) are excluded. The non-single-molecule signal is
likely caused by aggregation of dsDNA or probe ssDNA. In the main text we
presented the transcription detection trajectories, in which the mRNA channel
showed signal burst. We also found there exist trajectories that in the mRNA
channel the signal showed single-molecule level intensity (>1000 Hz), but it per-
sisted throughout the duration of the trajectory, as showed in Fig. S21 (a–c).
Furthermore, trajectories exist in which the mRNA channel showed increasing
signal, above single-molecule level, persisting until the DNA was no longer tracked
as showed in Fig. S21(d–f). These results may be caused by the probe DNA
unspecific binding to the dsDNA.

**Alexa 488 photo-bleaching time measurement.** The 1388 bp dsDNA labeled
with Alexa 488 was used to measure the photo-bleaching time of Alexa 488. The
1388 bp dsDNA was synthesized by PCR using Primer 2 and Primer 5, and pT7-
Spi2-8R as the template.

Primer 5: 5′-/Amino/ TTT CCA CAA CGG TTT CCC TCT AG.
Primer 5 was labeled with Alexa Fluor 488 NHS using the same labeling
procedure as described above, prior to PCR. 12 pM DNA was dissolved in PBS
solution and then 500 μL of solution was applied to a quartz coverslip before the
experiment. The DNA can bind to the coverslip non-specificity. In order to
locate the dye in the center of laser volume, the feedback tracking was turned on
at the beginning of measurement. Once the dye was captured, the feedback
tracking was turned off and the laser will continually illuminate on the dye
molecule. The used laser power is the same with that in the transcription
tracking experiment.

**Precision measurement.** 1388 bp dsDNA labeled with Atto 647 N was used to
measure the tracking precision (Supplementary Fig. 24). The 1388 bp dsDNA was
synthesized by PCR using Primer 2 and Primer 5, and pT7-Spi2-8R as the tem-
plate. Primer 2 was labeled with Atto 647 N NHS using the same labeling proce-
dure as described above, prior to PCR. 2 pM DNA was dissolved in PBS solution
and then 500 μL solution was applied to a quartz coverslip before the experiment.
The tracking precision was calculated with the standard deviation of each axis
position of 1 s duration.

**Reporting summary**. Further information on research design is available in the Nature Research Reporting Summary linked to this article.

## Data availability

The data that support the findings of this study are available from the corresponding author upon reasonable request.

## Code availability

Code used for data analysis is available at https://github.com/welsherlab/3DSMART

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

## Acknowledgements

The authors would like to thank Prof. Haw Yang for helpful discussions. The authors acknowledge financial support from the National Institute of General Medical Sciences of the National Institutes of Health under award number R35GM124868, the National Science Foundation under Grant No. 1847899, and from Duke University.

## Author contributions

S.H., J.E., and K.W. conceived the idea. SH and J.E. performed the experiments. S.H., J.E., and K.W. analyzed the data and wrote the manuscript.

## Competing interests

The authors declare no competing interests.

## Additional information

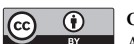

