## [Peer Review File · Nature Communications]

REVIEWERS' COMMENTS:

Reviewer #1 (Remarks to the Author):

The authors have addressed all of our concerns. I believe that the manuscript reports significant advancement in single-molecule tracking technology, pioneered by (Berg, Gratton, Mabuchi, Moerner, Cohen, Yang, Werner and others). Therefore, I fully support its publication in the current form.

Reviewer #3 (Remarks to the Author):

"Real-time 3D Single Molecule Tracking" by Huo et al. presents a novel 3D single-molecule active real-time tracking method (3D-SMART) which tracks single dye molecules in solution for *minutes* at a time with photon-limited temporal resolution. The impact of this paper lies in the new approach to tracking, which uses active feedback to lock-in on molecules for this long duration, and which is sensitive enough to capture even fairly fast ($D \sim 10 \mu\text{m}^2/\text{s}$) and fairly dim (single Atto 647N fluorophores) targets. I definitely support publication of this manuscript in Nature Communications and believe that the community will find much utility in 3D-SMART.

The long duration of tracking is quite impressive. In response to the comments of reviewer 2, the authors have clarified the text in a way that in fact begins to highlight these advantages. For instance (p. 6) "It should be noted that the long tracking duration shown here does not rely on the oxygen scavenger system. This may be due in part to the fact that the molecule is only periodically illuminated in this scan pattern, allowing for dark-state relaxation". I am very satisfied with the reasoning that the authors use to address these questions, and in fact, this response shows that the system is going to be widely applicable, for instance for subcellular imaging – if on the other hand, an oxygen scavenger was required, then live-cell experiments would be precluded! Though the text is fine as is, I would recommend that the authors actually add even more text or at least create a separate paragraph on page 6 to really highlight the advantages of the periodic scan over constant wide-field illumination – for instance in the response to reviewer 2, the authors note

[REDACTED]

– if this statement could be backed up with simple in vitro measurements or a reference to the literature, that would be very relevant here.

I also appreciate the changes that have been made in response to reviewer 2 to add more detail to how the bursts were recognized.

Minor comments: Based on the units, $10 \mu\text{m}^2/\text{s}$ in the abstract is a "diffusion coefficient" not a "speed". Indeed, throughout the paper, I would recommend changing "diffusive speed" to "diffusion coefficient".

REVIEWERS' COMMENTS:

Reviewer #1 (Remarks to the Author):

The authors have addressed all of our concerns. I believe that the manuscript reports significant advancement in single-molecule tracking technology, pioneered by (Berg, Gratton, Mabuchi, Moerner, Cohen, Yang, Werner and others). Therefore, I fully support its publication in the current form.

Response: Thanks for your support and we appreciate for your time.

Reviewer #3 (Remarks to the Author):

“Real-time 3D Single Molecule Tracking” by Huo et al. presents a novel 3D single-molecule active real-time tracking method (3D-SMART) which tracks single dye molecules in solution for *minutes* at a time with photon-limited temporal resolution. The impact of this paper lies in the new approach to tracking, which uses active feedback to lock-in on molecules for this long duration, and which is sensitive enough to capture even fairly fast ($D \sim 10 \mu\text{m}^2/\text{s}$) and fairly dim (single Atto 647N fluorophores) targets. I definitely support publication of this manuscript in Nature Communications and believe that the community will find much utility in 3D-SMART.

Response: Thanks for your support and we appreciate for your time.

The long duration of tracking is quite impressive. In response to the comments of reviewer 2, the authors have clarified the text in a way that in fact begins to highlight these advantages. For instance (p. 6) “It should be noted that the long tracking duration shown here does not rely on the oxygen scavenger system. This may be due in part to the fact that the molecule is only periodically illuminated in this scan pattern, allowing for dark-state relaxation”. I am very satisfied with the reasoning that the authors use to address these questions, and in fact, this response shows that the system is going to be widely applicable, for instance for subcellular imaging – if on the other hand, an oxygen scavenger was required, then live-cell experiments would be precluded! Though the text is fine as is, I would recommend that the authors actually add even more text or at least create a separate paragraph on page 6 to really highlight the advantages of the periodic scan over constant wide-field illumination – for instance in the response to reviewer 2, the authors note **[REDACTED]**

– if this statement could be backed up with simple in vitro measurements or a reference to the literature, that would be very relevant here.

Response: Thanks for your suggestion. We have added more text on the mechanism of periodic scan reduced photobleaching. We also cited several literatures supported this theory.

Revise: "It should be noted that the long tracking duration shown here does not rely on an oxygen scavenger system. This may be due in part to the fact that the molecule is only periodically illuminated in this scan pattern. It has been observed that the major photobleaching pathway in fluorophores results from photoexcitation of long-lived triplet states³⁵⁻³⁷. As a result, avoiding the excitation and allowing relaxation of the triplet state can reduce the rate of photobleaching. Pulse optimization to allow for dark state relaxation has led to increased photon yields in both confocal laser scanning^{38,39} and STED microscopies^{40,41}. The fast, periodic illumination pattern of 3D-SMART likely reduces photobleaching in a similar way, where the relatively long time intervals between illumination of the tracked molecule permits triplet relaxation."

I also appreciate the changes that have been made in response to reviewer 2 to add more detail to how the bursts were recognized.

Response: Thanks.

Minor comments: Based on the units, 10 $\mu\text{m}^2/\text{s}$ in the abstract is a "diffusion coefficient" not a "speed". Indeed, throughout the paper, I would recommend changing "diffusive speed" to "diffusion coefficient"

Response: Thanks for your suggestion and we have modified 'diffusive speed' to 'diffusion coefficient' throughout the manuscript.